# Analyzing Online Fake News Using Latent Semantic Analysis: Case of USA Election Campaign

**Richard G. Mayopu, Yi-Yun Wang and Long-Sheng Chen \***

Department of Information Management, Chaoyang University of Technology, Taichung 413310, Taiwan
* Correspondence: lschen@cyut.edu.tw

**Abstract:** Recent studies have indicated that fake news is always produced to manipulate readers and that it spreads very fast and brings great damage to human society through social media. From the available literature, most studies focused on fake news detection and identification and fake news sentiment analysis using machine learning or deep learning techniques. However, relatively few researchers have paid attention to fake news analysis. This is especially true for fake political news. Unlike other published works which built fake news detection models from computer scientists' viewpoints, this study aims to develop an effective method that combines natural language processing (NLP) and latent semantic analysis (LSA) using singular value decomposition (SVD) techniques to help social scientists to analyze fake news for discovering the exact elements. In addition, the authors analyze the characteristics of true news and fake news. A real case from the USA election campaign in 2016 is employed to demonstrate the effectiveness of our methods. The experimental results could give useful suggestions to future researchers to distinguish fake news. This study finds the five concepts extracted from LSA and that they are representative of political fake news during the election.

**Keywords:** online news; fake news; latent semantic analysis; natural language processing; social media

## 1. Introduction

Fake news has become a scourge in democratic countries such as Indonesia, Kenya, Lebanon, and many other democratic countries in the world over the past decade. People in the world have had to face the situation since 2012 [1] when hoaxes and disinformation caused distractions in many countries. Fake news can be defined by perceptions of real or genuine news. By knowing these characteristics, one can easily tell what is fake and what is real.

According to experts, fake news is currently the way to gain sympathy through social media and/or any kind of internet functionality [2]. Some fake news spreads through social media and private messaging apps. Social media can have both positive and negative benefits. Social media can be the worst channel for wreaking social havoc. Drawing lessons from the 2016 US presidential election and most recently the 2020 US presidential election [3], many researchers have found that fake news plays a role in the game, especially in developing countries such as Indonesia, Myanmar, Thailand, and Malaysia. Sadly, the most cases of fake news were found in Southeast Asia. However, the United States is a big, developed and democratic country that affects the influence of spreading fake news during the presidential election.

Technology plays a role in the regulation of alternative communication media and mass media. If traditional mass media dominates the industry, today, alternative media such as the Internet will benefit from this situation. The integration of mass media has become a hot topic in mass media research and a new form of network news. Others consume digital news or news on digital platforms [4]. Although there are many terms,

in this article we will discuss online news. Online news has become an easy solution to obtaining information without reading the newspaper. Several online news sources provide free access for readers because the cost of building online news is not expensive and difficult if we are comparing it with the newspaper. On the other hand, online news also gives the potential to produce fake information. While people love to read online news because of the convenience of online platforms, they are also afraid of the impact of communication. Furthermore, in the disruption of the information era, fake news can also be used to manipulate public opinion and create division and unrest. By spreading false information, individuals or groups with a particular agenda can create a sense of fear or anger among certain populations, leading to social unrest or even violence.

Most published studies discuss how to detect fake news [5]. In terms of algorithms, there are many that can be used, depending on the object of detection [6]. Instead of understanding algorithms, social scientists, especially communication scientists, have opted for artificial methods to detect fake news. Meanwhile, IT scientists and professionals are trying to figure out the best way to detect fake news.

In fact, many social scientists have studied fake news using classic methods such as interviews, observations, and questionnaires [7,8]. This is how limited data and small-scale fake news are collected. Additionally, this kind of research is only for traditional newspapers or broadsheets. However, in modern times, especially on online platforms, fake news is easy to spot in the digital age. The amount of fake news has increased dramatically. Facing the era of big data, it is becoming increasingly difficult to analyze fake news using traditional methods used by social scientists. Therefore, social scientists need effective methods to help them achieve their goals.

In recent years, research in artificial intelligence (AI) has been considered the best solution to find the best results in detecting fake online news. A popular technique used in fake news research is natural language processing (NLP) using text mining methods. To obtain results based on previous research purposes, deep learning models such as CNN (convolutional neural network), RNN (recurrent neural network), BERT (bidirectional encoder representations from transformers), and LSTM are generally popular [9]. According to the existing literature, regarding fake news, most of the research focuses on fake news detection and recognition [10–12] and fake news sentiment analysis [13–15], but relatively few researchers have focused on fake news analysis [16–18]. This is especially true for fake political news. Additionally, many researchers using artificial intelligence techniques to detect fake news are only doing so from a computer science perspective.

Therefore, unlike other works that build fake news detection models for computer scientists' opinions, this study aims to develop an effective method that combines natural language processing (NLP) and latent semantic analysis (LSA) using singular value decomposition (SVD) techniques to help social scientists discover the exact elements by using analysis of fake news. In addition, the authors analyze the characteristics of real news and fake news. The effectiveness of this research approach is demonstrated with a real-life example from the 2016 US election. The experimental results can also provide useful suggestions for future researchers to distinguish fake news.

## 2. Related Works

### 2.1. The Ideologies of Journalism

In a common definition, journalism is the activity or practice of gathering facts and turning them into information for communication to society or the public. There are four ideologies in journalism that affect the process of the news report, which are described as follows: authoritarianism [19], liberalism [20], Soviet communism [21] and social doctrine [22]. Scholars' research over the years has shown that the influence of journalistic ideologies is increasing, but grand ideologies remain. There are four main ideologies that usually implement new concepts such as digital journalism, digital public relations, digital journalism, etc. [23].

Journalism, public relations, and digital media are areas of communication and media studies covered by the journalism studies subject. The aim is usually to improve methods on fake news topics. It is clear that researchers have documented the history of the development of the concept of news [24]. Regarding fake news research, our study evaluates key variables such as news, progress, method, media type, media platform, and the most commonly used terms in fake news.

Previous studies have used fake news theory, described the history of fake news, and attempted to explain it in new ways, showing that fake news is diverse in journalism. Since previous pilot tests only found a few publications on fake news from before the 2000s, analysis focused on peer-reviewed research published by publishers. The rise of digital and social media in the early 2000s coincided with the spread of the fake news phenomenon. Based on the development of news, fake news has also changed to adapt to new types of news. This is referred to in the study as a new generation of fake news. The study provides an empirical review of the research discussed, progress, theories, methodologies, media types and platforms, most commonly used words, and the distribution of fake news studies [25].

*2.2. Online News, Online Fake News and Social Media*

In fact, the definition of online news is similar to that of traditional news or print news. The difference is the platform used to deliver and distribute the news. However, we must take into account that its conceptual development cannot be ignored. Although, it has similar definitions, web news still has unique parts to describe it. Online news is fact-based information delivered through the use of technology platforms such as websites and mobile website applications, and also via social media (the most popular). As far as social media is concerned, there are many varieties and processes of delivering news, but what matters is which social media is popular for consumers to use.

The fact of digitization and the fact that all text (symbolic meaning in all coded and recorded forms) can be reduced to binary code and share the same production, distribution and storage processes are perhaps the most fundamental aspects of information and communication technology (ICT) [26]. In fact, the advent of online journalism coincided with a great time for the news age. While the infrastructure was designed for television and radio, the internet adopted rules to spur the rise of online news. In Indonesia, the first online news was detik.com. This is a company that provides information on the country, but as a common point in Indonesian mass media, the dominant form of information is all about the capital city, "Jakarta". However, online news knows no borders, which means that people in other places or countries can obtain information in real time.

A previous study tried to define "fake news" [27]. This study implements this concept using the underlying theory and diversity of journalism. It provides a concept of news in social media, and a typology of news. There are concepts of typological media such as satirical journalism, parody journalism, fabricated journalism, photo manipulation, public relations and publicity in advertising and journalism. The study builds on an examination of how the term "fake news" has been defined and implemented in previous studies. An analysis of 34 academic articles published between 2003 and 2017 that used the term "fake news" yielded a taxonomy of fake news types: news satire, news parody, fabrication, manipulation, advertising, and propaganda. These criteria are based on two dimensions: the level of fact and the level of deception.

Fake news has been defined by other scholars, and based on these definitions, a framework is provided to conceptualize the different types of fake news identified in the literature. A study identified a typology of fake news definitions guided by areas of authenticity and intent. Satire, parody, and fabrication are types of fake news, and new recommendations reveal that misinformation, disinformation, clickbait, rumors, and propaganda [28] can be identified as fake news, although this remains under debate among researchers. Therefore, according to the new concept, we must carefully identify the kind of news. A clear definition of fake news, one that corresponds to its empirical expression, helps

to test and develop theories of news creation and consumption. However, the typology created depends only on how past academic researchers have defined the term. News satire, for example, constitutes fake news, at least in the way those concepts are defined. Currently, journalists are forced to separate, if not defend, their work from fake news.

*2.3. Mainstream Media Spread the Fake News*

At the beginning of this section, we better understand the impact of fake news. Fake news has a certain impact and contributes to civilization, but society also faces a difficult situation in distinguishing between fake news produced by online mainstream media and online fake media. Tsfati et al. [29] found "consequences of mainstream media spreading fake news". According to the fake news site, their findings only affected a small portion of the public. Therefore, on the other hand, the data suggests that a significant portion of people know about notable fake news stories and believe them. Mainstream media or official news organizations produce and are responsible for sharing news that may be part of the spread of fake news. The mainstream media on the Internet plays a role in the dissemination of fake news, the reasons for reporting it, and its impact on audiences. Therefore, major news media play an important and critical role in the dissemination of fake news. While there are no empirical estimates of how many people are exposed to fake news stories through mainstream news outlets, research suggests a concentration and relatively limited exposure to fake news on social media, as well as the fact that some stories are remembered, endorsed, and even a large portion of the audience believe them. This strongly suggests that, at least when it comes to the most reported stories, more people learn about them from mainstream news outlets. As the evidence is circumstantial, though it relies on reasoning rather than direct evidence, it is very convincing.

Mainstream media should provide evidence that they have a high degree of credibility in disseminating information through their type of journalism. According to the type of fake news exposure, the exposure of online fake news is different in five countries: Lebanon, Qatar, Saudi Arabia, Tunisia, and the United Arab Emirates [30]. This comes at the same time as political momentum increases with election events, the socialization of public policy from politicians to society, etc. The study of those countries understandably uses the term fake political news exposure (FNE). The mainstream media, sometimes carelessly spreads disinformation. However, mainstream media also often provides incomplete information and passes it on to the public. This is the worst journalistic practice, as news must be delivered to the public in its entirety and verified. Journalistic credibility is a concept that has involved journalism in the past. Today, the public is also considering the credibility and reputation of mainstream media.

*2.4. Fake News Detection*

In related works, most of the published studies have paid attention to fake news detection. For examples, in the work of Zhang et al. [12], they presented a computational approach which leverages event and topic extraction techniques coupled with a topic-merging mechanism to process news data and reduce the number of topics required to detect fake news in a real-time manner. Capuano et al. [10] found a content-based fake news detection model which can perform excellently over multiple datasets/topics. In order to improve the shortcomings of existing propagation-based fake news detection algorithms, Song et al. [11] presented a dynamic propagation graph-based fake news detection method to classify fake news.

In addition to studies on fake news detection, there are studies focusing on the sentiment classification of fake news. For instance, Iwendi et al. [13] used deep learning models to detect COVID-19-related misinformation on social media. Pratama and Tjahyanto [15] used machine learning approaches, including support vector machine (SVM) and Naïve Bayes (NB) to study the influence of fake accounts on sentiment analysis. For identifying fake or harmful information, Lin et al. [14] employed a bidirectional encoder representation from transformers (BERT)-based model which applies ensemble learning methods with

text sentiment classification. To sum up, from the available literature, most of the studies focus on fake news detection and fake news sentiment analysis using machine learning or deep learning methods. However, relatively few researchers have paid attention to fake news analysis. Consequently, this study aims to analyze fake news analysis.

*2.5. Latent Semantic Analysis*

LSA is more powerful at extracting meaningful concepts within semantic meaning [31]. Through LSA, the text data (TDM) are subjected to singular value decomposition (SVD) to construct a semantic vector space that can be used to represent the concept–word–document association [32,33]. LSA has been applied successfully in diverse language systems for computing the semantic similarity of texts [34]. Therefore, LSA has been successfully applied to discover useful knowledge from a huge number of documents. For example, Chen at el. [33] utilized K-means and LSA to organize the selected keywords into understandable concepts in live streaming. In the work of Hsiao and Hsiao [35], the authors also used LSA to find hotel quality elements from online reviews. In order to understand fake news concepts, this study also employed LSA.

## 3. Methodology

Previous research usually used machine learning or deep learning techniques to identify fake news. However, the main purpose of this study is to analyze the components of fake news. Consequently, latent semantic analysis (LSA) was employed to extract significant sentences from an input document in order to create a summary by identifying a latent or hidden semantic structure [36]. When analyzing collected documents, the LSA method uses singular value decomposition (SVD) to derive concepts from collected documents by modeling them as a term–document matrix (TDM). The process of this study can be found in Figure 1. The detailed steps of the implemental procedure are given below.

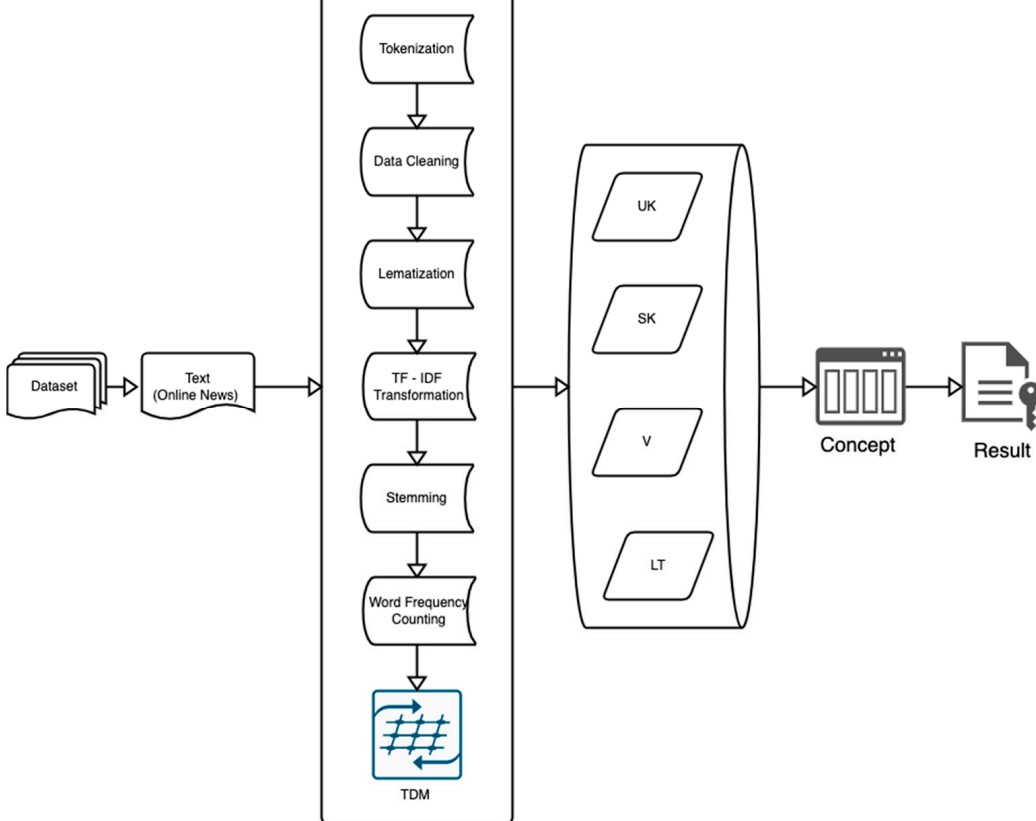

**Figure 1.** The implemental process of this study.

The technical parameters of the implementation are described as follows:

Hardware: Processor Intel® Core ™ i7-4790 CPU © 3.60 GHz (8Cpus), ~3.6 GHz; Memory: 32768MB RAM; System Manufacturer: Hewlett-Packard.
Software: Anaconda Jupiter Notebook 6.5.4 for obtaining the TF–IDF matrix and MATLAB R2022a for the LSA-SVD process; Operating System: Windows 10 Pro, 64 bit (10.0 Build 19044).

*Step 1. Dataset collection*

First of all, according to the needs of our experiment, the online news data were collected. This study takes election fake news as the research object. The analysis is based on a special collection of online news data gathered for the purpose of investigating fake news during election campaigns. This dataset is not representative of other types of fake news or themes; therefore, this study is strongly applicable to analyzing the fake news of the political election.

*Step 2. Data pre-processing and NLP*

Python was used to pre-process the data in order to examine the data set that could be used in this study. The pre-processing stage entailed a number of steps, such as tokenization and lemmatization, before obtaining the TDM for LSA analysis. In fact, NLP frequently takes into account the intriguing text mining applications, and this method is qualified to extract pre-existing knowledge from the text. The complex sentences that occasionally appear in tweets with the intention of objective communication can be significantly extracted using NLP [37]. The implementation process can be summarized as follows.
Step 2.1: Tokenization
The Python Natural Language Toolkit (NLTK) was used for text tokenization. This study employed "unigram" to segment sentences.
Step 2.2: Data cleaning
In this step, we eliminated stop words such as "the" and "or", non-English words, and meaningless icons to obtain clean data for further analysis.
Step 2.3: Lemmatization
This step involves reducing complex forms of a single word to their most basic form, such as "ate" and "eaten" to "eat".
Step 2.4: Word Frequency Counting
The authors performed a word frequency count and eliminate words with a frequency of less than 5.
Step 2.5: Constructing a Term–Document Matrix (TDM)
In this step, the authors created a term–document matrix (TDM) using the TF–IDF (term frequency–inverse document frequency) weights shown in Equation (1).

$$TF - IDF = TF(t_i, d_i) \times \log\left(\frac{N}{N(t_i)}\right) \tag{1}$$

where $t_i$ is the $i$th term; $d_i$ represents the $j$th document; $N$ is the total number of all documents; $N(t_i)$ denotes the number of documents which contain $t_i$ features.

*Step 3: Latent Semantic Analysis (LSA)*

The relationship between words and their concepts was examined using singular value decomposition (SVD).
Step 3.1: SVD
In Figure 2, A represents TDM. Three matrices are produced by the SVD function "terms matrix $(U_{t \times r})$", "orthogonal matrix $(S_{r \times r})$", and "document matrix $(V_{r \times n}^T)$", where $t$ refers to the number of words, $n$ refers to document term, and $r$ refers to the number of concepts in the semantic space. Figure 2 shows the SVD process.

$$A_{(t \times n)} = U_{(t \times r)} \times S_{(r \times r)} \times V^T_{(r \times n)}$$

**Figure 2.** Singular value decomposition.

Step 3.2: Dimension Reduction

A lot of unimportant information may still be contained after running SVD, so it is necessary to reduce the dimensional space. It is crucial to choose the feature value *k* in order to preserve the original characteristics. The scree test is used in this study to calculate the *k* value. A scree plot always displays the eigenvalues in a downward curve, ordering the eigenvalues from largest to smallest. According to the scree test, the "elbow" of the graph where the eigenvalues seem to level off is found and concepts to the left of this point should be retained as significant. In Figure 3, since we choose *k* concepts, we only need to take the $S_k$ matrix, which lowers the dimensionality.

$$A_{k\,(t \times n)} = U_{k\,(t \times r)} \times S_{k\,(r \times r)} \times V^T_{(r \times n)}$$

**Figure 3.** SVD after dimension reduction.

Step 3.3: Orthogonal rotation of axes

The concept load ($L_T$) is then calculated by multiplying the dimensionally delimited concept $U_k$ with the concept $S_k$, which is calculated using Equation (2) to obtain the concept load $L_T$, each feature word is ranked according to the load, and the word concept is named.

The dimensionally delimited concept ($U_k$) is multiplied by the concept $S_k$, which is calculated using Equation (2), to obtain at the concept load ($L_T$). Each feature word is then ranked according to the load ($L_T$), and the word concept is named.

$$L_T = U_k \times S_k \qquad (2)$$

*Step 4: Naming the extracted concepts*

In this step, the authors focused on naming the concepts extracted in the next steps. The name of concepts were based on the keywords that were extracted in the previous step. These extracted meaningful concepts, furthermore, were analyzed and contributed to the understanding and definition of every single concept.

*Step 5: Explaining Results*

The authors then explained the result of the LSA, which named the concepts in Step 4. Moreover, the analysis of accurate and fake news was provided based on our finding.

*Step 6: Drawing discussions and conclusions*

The authors drew discussions and conclusions based on the results. The discussion section contains four parts as described. First, it focuses on how to distinguish and analyze

accurate and fake news. Second, in order to identify fake news, we analyze the structure of accurate news and fake news. Third, we discuss the type of fake news that usually gives delayed facts or incomplete facts. Fourth, the impact of fake news in the disruption era and in the political situation is very important to be discussed. Finally, a conclusion is made.

## 4. Experimental Results

*Results of LSA for Fake News during USA Election Campaign*

In this study, the experiment was conducted by using python in order to obtain the TDM from the dataset to be analyzed using MATLAB. The user data, which included fake news during USA's presidential election campaign in the year 2016, came from Kaggle (https://www.kaggle.com/code/therealsampat/fake-news-detection/data (accessed on 8 June 2022)), and namely included "fake news detection". In total, this dataset contains 17,903 news sources from March 2016 to December 2017 with a title, text, subject, and date. Among them, only 5009 fake news examples with the "text" variable were left for further analysis. The reason for selecting this dataset was because this was the political campaign period for the US presidential election. Table 1 shows a sample of the used fake news. In this table, lots of meaningless icons in the original data appear. The implementation of the NLP process is described in Step 2. Finally, we construct a 5009 × 10,019 TDM with TF–IDF weights for LSA.

**Table 1.** An example of used fake news.

| Title | Text | Subject | Date |
|---|---|---|---|
| Sheriff David Clarke Becomes An Internet Joke For Threatening To Poke People In The Eye | On Friday, it was revealed that former Milwaukee Sheriff David Clarke, who was being considered for Homeland Security Secretary in Donald Trump s administration, has an email scandal of his own. In January, there was a brief run-in on a plane between Clarke and fellow passenger Dan Black, who he later had detained by the police for no reason whatsoever, except that maybe his feelings were hurt. Clarke messaged the police to stop Black after he deplaned, and now, a search warrant has been executed by the FBI to see the exchanges. Clarke is calling it fake news even though copies of the search warrant are on the Internet. I am UNINTIMIDATED by lib media attempts to smear and discredit me with their FAKE NEWS reports designed to silence me, the former sheriff tweeted. I will continue to poke them in the eye with a sharp stick and bitch slap these scum bags til they get it. I have been attacked by better people than them #MAGA I am UNINTIMIDATED by lib media attempts to smear and discredit me with their FAKE NEWS reports designed to silence me. I will continue to poke them in the eye with a sharp stick and bitch slap these scum bags til they get it. I have been attacked by better people than them #MAGA pic.twitter.com/XtZW5PdU2b David A. Clarke, Jr. (@SheriffClarke) 30 December 2017 He didn t stop there. BREAKING NEWS! When LYING LIB MEDIA makes up FAKE NEWS to smear me, the ANTIDOTE is go right at them. Punch them in the nose & MAKE THEM TASTE THEIR OWN BLOOD. Nothing gets a bully like LYING LIB MEDIA S attention better than to give them a taste of their own blood #neverbackdown pic.twitter.com/T2NY2psHCR David A. Clarke, Jr. (@SheriffClarke) 30 December 2017 The internet called him out. This is your local newspaper and that search warrant is not fake, and just because the chose not to file charges at the time does not mean they will not! Especially if you continue to lie. Months after decision not to charge Clarke, email search warrant filed https://t.co/zcbyc4Wp5b KeithLeBlanc (@KeithLeBlanc63) 30 December 2017 I just hope the rest of the Village People are not implicated. Kirk Ketchum (@kirkketchum) 30 December 2017 Slaw, baked potatoes, or French fries? pic.twitter.com/fWfXsZupxy ALT-Immigration (@ALT_uscis) 30 December 2017 pic.twitter.com/ymsOBLjfxU Pendulum Swinger (@PendulumSwngr) 30 December 2017 you called your police friends to stand up for you when someone made fun of your hat Chris Jackson (@ChrisCJackson) 30 December 2017 Is it me, with this masterful pshop of your hat, which I seem to never tire of. I think it's the steely resolve in your one visible eye pic.twitter.com/dWr5k8ZEZV Chris Mohney (@chrismohney) 30 December 2017 Are you indicating with your fingers how many people died in your jail? I think you're a few fingers short, dipshit Ike Barinholtz (@ikebarinholtz) 30 December 2017 ROFL. Internet tough guy with fake flair. pic.twitter.com/ulCFddhkdy KellMeCrazy (@Kel_MoonFace) 30 December 2017 You re so edgy, buddy. Mrs. SMH (@MRSSMH2) 30 December 2017 Is his break over at Applebees? Aaron (@feltrrr2) 30 December 2017 Are you trying to earn your still relevant badge? CircusRebel (@CircusDrew) 30 December 2017 make sure to hydrate, drink lots of water. It's rumored that prisoners can be denied water by prison officials. Robert Klinc (@RobertKlinc1) 30 December 2017 Terrill Thomas, the 38-year-old black man who died of thirst in Clarke s Milwaukee County Jail cell this April, was a victim of homicide. We just thought we should point that out. It cannot be repeated enough. Photo by Spencer Platt/Getty Images. | News | 30 December 2017 |

Next, we identified the k value. Figure 3 shows the scree plot. According to Figure 4, the k-value should be determined as carried out in Section 5.

**Figure 4.** Scree plot.

The concept was retrieved from the data during the SVD process and the result found the concepts with the highest loadings. The authors then identified four keywords in each loading and furthermore compiled the keywords into the precision concepts. Five concepts were found. Table 2 lists the top ranked keywords and their loadings of each concept. Next, according to the representative keywords in a specific concept, we could name these extracted concepts. Table 3 summarizes the naming of the concepts and their representative glossary.

**Table 2.** Extracted keywords and loadings in each concept.

| Concept #1 | | Concept #2 | | Concept #3 | |
|---|---|---|---|---|---|
| Keywords | Loadings | Keywords | Loadings | Keywords | Loadings |
| Political | 0.3745 | Senator | 0.2007 | Fact | 0.3739 |
| Administration | 0.3524 | Political | 0.1552 | Great | 0.2840 |
| Much | 0.2731 | Election | 0.1493 | Justice | 0.1549 |
| America | 0.2466 | Campaign | 0.1147 | Life | 0.1124 |
| Republican | 0.1910 | Republican | 0.0943 | public | 0.0839 |
| senate | 0.1901 | Clinton | 0.0750 | message | 0.0835 |
| election | 0.1866 | senate | 0.0736 | idea | 0.0783 |
| breaking | 0.1857 | Donald | 0.0697 | medium | 0.0779 |
| accused | 0.1692 | message | 0.0674 | time | 0.0751 |
| justice | 0.1659 | reality | 0.0610 | America | 0.0670 |
| … | … | … | … | … | … |

| Concept #4 | | Concept #5 | |
|---|---|---|---|
| Tweeted | 0.2679 | Report | 0.2151 |
| Clinton | 0.2297 | Pic | 0.1585 |
| Talking | 0.1095 | Campaign | 0.1486 |
| Say | 0.1011 | Medium | 0.1343 |
| claim | 0.0897 | world | 0.1090 |
| decided | 0.0876 | American | 0.0641 |
| great | 0.0714 | United | 0.0588 |
| medium | 0.0671 | FBI | 0.0552 |
| justice | 0.0559 | senate | 0.0547 |
| twitter | 0.0541 | Russia | 0.0540 |
| … | … | … | … |

**Table 3.** Extracted concepts.

| No. | Concept Name | Representative Glossary |
|-----|--------------|------------------------|
| 1 | Coalition | Political, Administration, Much, America |
| 2 | Politic | Senator, Political, Election, Campaign |
| 3 | Future | Fact, Great, Justice, Life |
| 4 | Statement | Tweeted, Clinton, Talking, Say |
| 5 | Issues | Report, Pic, Campaign, Medium |

The concepts were built based on the LSA results. However, the issue of creating coherent stories out of the concepts by combining keywords with related meanings inevitably came up. The terms with similar meanings found here were actually contextual similarities based on the frequency of co-occurrence within documents, similarly to the findings from Wang et al., and did not imply the concept of synonyms [38]. In other words, two terms in a group were said to be highly similar if they appeared in a variety of contexts describing the same subjects. For instance, 'Political', 'Administration', 'Much', and 'America' were all grouped together. Next, we explain the extracted concepts of fake news.

Concept #1: Coalition

The concept of the representative glossary containing 'Political', 'Administration', 'Much', and 'America' is named "coalition". Coalition is a part of political practice, and it is needed by politicians and or parties. Moreover, coalition is usually used in order to obtain power in political activity. On the other hand, coalition is only temporary because the relationship among parties is not a long-term relationship [39]. It depends on the goals of the parties during the campaign and election and is also for the contender to run the government once they become the winner of the election.

Concept #2: Politic

For the glossary containing 'Senator', 'Political', 'Election', and 'Campaign', the concept is "politic". Politic is the virtue of influencing people to do good things for the good of civilization. Researchers also define politic based on two perspectives: connotation and denotation. In the connotation perspective, 'politic' is considered a bad way to influence people in certain groups such as tribes or ethnicities; we might define these as groups of interest. On the other hand, the denotation perspective provides the good meaning of 'politic'. Politic is the responsibility of managing resources such as by educating people, protecting natural resources, and developing policies for providing welfare to society. Moreover, the meaning of 'political' implies the cooperation of the government with senate activity to produce good policy.

Concept #3: Future

From the representative glossary containing 'Fact', 'Great', 'Justice', and 'Life', the concept is "future". 'Future' is hope and people living with hope for the future. Hope possibly drives people to invest in a better life for the future. In the political environment, hope can be plenty, particularly in coalitions formed to build good relationships. Moreover, 'future' is the things that are expected to happen in upcoming times.

Concept #4: Statement

In this concept, the representative glossary contains 'Tweeted', 'Clinton', 'Talking', and 'Say'. We can call the concept "statement". 'Statement' represents the crucial communications in politics. This is a kind of official communication from politicians or a very important person (VIP) to public. There are two types of 'statement', which are speaking and writing. In general, a statement, also known as an official statement, refers to very important messages that are delivered to the public. Official statements are usually delivered via mass media; however, nowadays, it usually delivered via social media. VIPs usually have a social media account in order to communicate with the public. However, the critical

issues include the fact that these accounts are not operated by the VIPs themselves. They have assistances to operate their accounts.

Concept #5: Issues

In this concept, the top glossary terms are 'Report', 'Pic', 'Campaign', and 'Medium'. We can name this concept "issues". Issues are rumors that have not been proven with facts. This terminology represents the noise in a political environment. However, issues, for some reason, are usually used in campaign activity. Therefore, issues must not be trustable until they can be proven and the true story confirmed. Mass media have also become the "big fans" of rumors, as they sometimes report issues through the news, potentially creating fake news because facts are ignored.

## 5. Discussions

### 5.1. Distinguishing the Difference between Fake News and Accurate News

From our results, five concepts of fake news were found and named. They are "coalition", "politic", "future", "statement", and "issues". As we know, these categories of news are usually employed to manipulate readers to influence the results of an election. Our findings can be used to classify the news which is fake that can influence readers with "precision" that is actually artificial and created by a fake news creator.

However, these concepts are only for fake news of the USA election campaign in 2016. We attempted to find components of fake news from these results. From the observation and the results of the LSA, we compared real news and fake news. Then, the differences between fake news and real news were recognized. The results are summarized in Table 4 and the table provides some of the differences. Firstly, fake news is based on social media and artificial fact, not based on fact. Secondly, fake news comes from a singular source, but real news is from multiple sources which usually cover both sides involved in the news. Thirdly, real news has both positive and negative viewpoints, but fake news has only one-sided information. Fourthly, identified and unidentified domains are websites or URLs that are used to publish or disseminate news or information that proves to be false or inaccurate. Additionally, credible news sources such as official news sites or media outlets can help identify areas associated with false or inaccurate news. In some cases, these news sources often provide official statements about fake news or information circulating on social media or other sites.

**Table 4.** Differences between fake news and accurate news.

| No. | Accurate News | Fake News |
|---|---|---|
| 1. | Based on fact/reality | Based on social media |
| 2. | Based on fact | Based on artificial fact |
| 3. | Contains many sources (covers both sides) | Contains a singular source |
| 4. | Contains the opposite information | Contains one-sided information |
| 5. | Contains an identified domain | Contains an unidentified domain |

Moreover, fake news employs some of the criteria of accurate news. Table 5 provides the criteria for newsworthiness from a journalism perspective. The following table summarizes seven elements of accurate news. However, after analysis, we can find that fake news can generally contain the same elements as true news. We give an example of true and fake news in Table 6. Additionally, we will analyze the structure of accurate and fake news.

**Table 5.** The criteria for distinguishing the systematics of accurate news and fake news.

| No. | Accurate and Fake News |
|---|---|
| 1. | 5 W + 1 H (what, when, why, where, who + How) |
| 2. | Structure (lead, body, and data) |
| 3. | Prominence |
| 4. | Proximity |
| 5. | Controversy |
| 6. | Impact |
| 7. | Human interest |

The structure of fake and real news can be analyzed based on Table 6. Both types of news have a complete structure, namely a title, lead and data section. Headlines are meant to display persuasive sentences, and sometimes both try to capitalize on the topic of the news. Additionally, in online news, whether true or false, it is common to use clickbait statements in the headlines to lure readers. Moreover, both have the ability to guide the description of news content. Fake news also uses various leads, namely the "What Lead", which contains information about what is going on or the "Who Lead", which contains information about who is involved in the incident." This thread also appears in accurate news. Accurate and fake news also both provide the body of the news, where more details of the events can be spelled out; however, we can also recognize that details in fake news can be wrong. In the last section, both also give some data to convince the reader that the news reported is very accurate. Even in the case of fake news, which feeds the wrong data, readers can be manipulated.

**Table 6.** The structure of accurate and fake news.

| Structure | Accurate News | Fake News |
|---|---|---|
| Title | Fox News's Shepard Smith debunks his network's favorite Hillary Clinton 'scandal,' infuriates viewers | WATCH: Delusional Trump Fans Lash Out At Fox Host For Reporting Facts Of Uranium One Deal |
| Lead | Fox News anchor Shepard Smith debunked what his own network has called the Hillary Clinton uranium "scandal," infuriating Fox viewers, some of whom suggested that he ought to work for CNN or MSNBC. | Clearly, Trump supporters want Fox News to lie to them. Because when Fox News host Shep Smith fact-checked Donald Trump s accusations against Hillary Clinton in regards to the Uranium One deal with Russia, they lost their shit. Smith thoroughly debunked Republican claims that Hillary Clinton approved the deal in a pay-to-play scam during her time as Secretary of State back in 2010 |
| Body | Smith's critique, which called President Trump's accusations against Clinton "inaccurate," was triggered by renewed calls from Republicans on Capitol Hill for a special counsel to investigate Clinton. Fox News, along with Trump and his allies, has been suggesting for months a link between donations to the Clinton Foundation and the approval of a deal by the State Department and the Obama administration allowing a Russian company to purchase a Canada-based mining group with operations in the United States. Trump called it "Watergate, modern-age". Former White House adviser Sebastian Gorka, speaking on Fox News last month, said it was "equivalent to" the Julius and Ethel Rosenberg spying case of the 1950s, in which the couple was charged with providing U.S. atomic secrets to the Soviet Union, noting that "those people got the chair". | First, the deal had to be approved by a committee of nine agency heads, who unanimously approved. Second, the State Department was represented by an assistant who says Clinton did not intervene. Third, the Uranium One deal stipulates that the uranium must be sold to civilian reactor operators in the United States, which blows Trump s claim that Hillary gave 20 percent of our uranium to Russia out of the water. Fourth, one man gave the Clinton Foundation all but $4 million of the $140 million donated by nine individuals associated with Uranium One. And that one man had already sold his stake in the company years before in 2007, well before Clinton even thought about being Secretary of State. And long before Barack Obama became president to make her Secretary of State. And, finally, Clinton had no power to veto or approve the deal herself. So, that means conservatives have no case against Clinton, effectively neutering any effort by the Justice Department to appoint a special prosecutor since doing so is contingent on the facts. And the facts support Clinton. So much so, that any court would laugh the obviously manufactured charges out of court. |
| Data | Various fact-checkers, including The Washington Post's, have already dismantled the underpinnings of these accusations. No one expected a similar debunking from Fox. But Smith, in his broadcast, made many of the same points as the fact-checkers. "Now, here's the accusation," he said. Nine people involved in the deal made donations to the Clinton Foundation totaling more than $140 million. In exchange, Secretary of State Clinton approved the sale to the Russians, a quid pro quo. The accusation [was] first made by Peter Schweizer, the senior editor-at-large of the website Breitbart in his 2015 book "Clinton Cash". The next year, candidate Donald Trump cited the accusation as an example of Clinton corruption. | Here's the video via YouTube. In response, Trump supporters threw a temper tantrum and called for Fox to fire Smith for reporting the facts. You need to get that hack @ShepNewsTeam out of Fox. He just made excuses for #CrookedHillary and is clearly a double agent. Hey Shep. Hillary took the 145M regardless of when it was sent. Pay for play you goof Lab Lover (@Dutchistheballs) 14 November 2017 |

### 5.2. Manipulation and Delayed Facts

It is a sensibility of journalistic practice that fake news is created to manipulate audiences [40]. In some cases, the term 'manipulation' refers to a Western perspective. For example, the spread of "fake news" is intricately linked to traditional (Western) media norms, which are themselves problematic. Subsequently, "fake news" is not a problem in itself, but a sign that the world media sphere needs to reimagine new normative approaches on the continent. Asia and Africa have their own norms, which may differ from those of Western continents such as Europe and Australia, as well as the US and Canada. In a post-Facebook–Cambridge Analytica world, this is an unacceptable position that fuels the flames of "fake news". Social media has become a concept that extracts the implementation of fake news, and is also a medium for disseminating fake news. The concepts presented in this article are not mutually exclusive, and they often interact in ways that are glossed over by the current "fake news" debate.

The discourse provides an outlet for the press to avoid any serious scrutiny of its crimes by placing the blame squarely on social media and social media users. Because of the intersection of these three elements, we must pay more attention to the news culture that exists in the African media industry, rather than engaging with the current popular topic of "fake news". Despite the structural differences in news consumption habits, the eloquently highlighted "information morality panic" has entered African discourse on "fake news".

Delayed facts are the least accepted concept when it comes to defining online news. Some journalists may disagree, while on the other hand some journalists do agree with the concept of delayed facts. The main feature of online news is its rapid release. This is also the main problem with lobbying messages. In trying to report the news, journalists often avoid the real facts. Therefore, the first time news is released, it always carries incomplete facts. In addition, journalists supplement the newly released news.

The delayed fact became an essential factor in its discovery in traditional newspapers. This contrasts with online news if newspapers usually present facts accurately. Either way, online news has more time to provide "continuous" facts because of its unlimited length and columns. Delayed facts are potentially able to distract online journalism from spreading valid information. They make readers confused while causing them to consume incomplete information.

### 5.3. The Impact of Fake News in the Disruption of the Information Era for the Current Situation

While research has recognized the impact of fake news, research has yet to systematically investigate the impact of fake news in an era of disruption. Typically, disruptive times are times when many intangible innovations emerge that the established organization, institution, company or agency is unaware of, so they disrupt the existing old system order and threaten to disrupt it. Today, people can become journalists known as citizen journalists; even if people do not have the capabilities of journalists, they can use social media platforms to gather information, edit it and share the information. The subversion of information is a "gift" of uncertain times. People can easily produce information that is not covered by mass media. This is the advantage of subversive information. On the other hand, people can also spread information that has not been spread by the mass media. With the journalistic standards we mentioned in the previous section of this article, this is a disaster for sharing information. Invalid information is a big problem for ordinary people who cannot understand how to verify information from potential fake news. This disinformation could turn into fake news and is now known as a hoax. The most obvious finding from the analysis is that information disruption can promote a good democracy if information-educated people can filter out and select good information to share. Mass media has a new responsibility to embed functionality. The fourth classical function of the mass media should be linked to the verification modality. Validation may be the central focus of the mass media in an age of information disruption. Information interruption is shown in Figure 5.

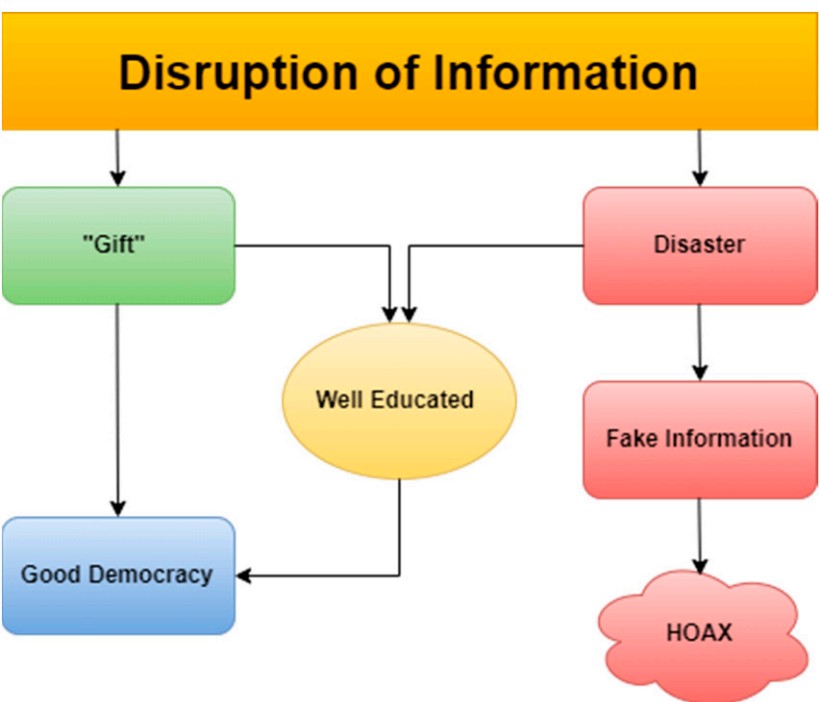

**Figure 5.** Fake news in the disruption of information era.

## 6. Limitation, Conclusions and Future Works

### 6.1. Limitation

The limitation of this research is specifically focused on political fake news, and narrows in on the situation of a presidential election that attracted great attention from fake news writers, which means that this research might be not reliable for different issues such as health, social matters, culture, finance, etc. Therefore, this research is not intended to make the generalization that this method can be useful for different type of news. Furthermore, SVD, used for the LSA study, also has limitations in terms of scalability and computational efficiency.

### 6.2. Conclusions and Future Works

This study aimed to analyze fake news by using LSA. Five important concepts of fake political news were discovered, including "coalition", "politic", "future", "statement", and "issues". These groups of concepts were used to create fake news to manipulate readers to influence the results of the election. To identify fake election news, these concepts could be used.

Moreover, analyzing the collected fake news in advance from a journalism perspective was also essential. The authors compared real news and fake news. Some significant differences could be found. First, fake news often comes from social media and fabricated facts. Second, fake news has a single source, while real news sources usually cover all sides of the news. Third, real news has both positive and negative points of view, while fake news has only one-sided information. Finally, fake news often comes from URLs with unidentified domains.

In addition, our research found the seven discriminant elements of the composition of accurate news and analyzed the article structures of true and false news. It was found that the components of true and false news are not different from the article structures. This means that fake news writers often masquerade fake news as accurate news. Therefore, it is impossible to distinguish the authenticity of news from the form of news composition and report article structure.

To sum up, this study combines NLP with LSA, effectively providing social scientists with an efficient approach to analyzing fake news. We find that LSA extracts five meaningful

concepts that can be used to effectively identify fake news. Additionally, from the point of view of journalism, it analyzes the structure of true and false news in detail. However, fake news will always evolve, producing better structures and sentences that readers cannot identify. The findings of this study contribute to our understanding of fake news in multiple ways and provide a basis for distinguishing fake news from accurate news. Incidentally, LSA is very well-established approach, and the used dataset is also published. However, the purpose of this study is to provide an easy tool for social scientists who typically use observational methods to describe what they see, rather than the more scientific data-based methods. Our proposed method can aid social scientists in discovering useful knowledge from the huge amount of fake news. Furthermore, the proposed method, although simple and understandable, has never been used in the field of analyzing fake news. This is also the main contribution of this study in practice.

In terms of the possible directions of future research, this study only used fake election news as the research object. In the future, researchers can try different topics. Perhaps we can have a deeper understanding of fake news and help news readers to judge fake news more easily. Additionally, we can stop the influence of fake news from affecting democratic societies in the information war. In addition, the used dataset only contains fake news. If readers can apply our method to identify factual news and then compare the results of this to the results of identifying fake news, this might be another interesting direction for future works.

**Author Contributions:** Conceptualization, R.G.M. and L.-S.C.; methodology, R.G.M. and Y.-Y.W.; software, R.G.M. and Y.-Y.W.; validation, R.G.M.; formal analysis, R.G.M. and L.-S.C.; writing—original draft preparation, R.G.M.; writing—review and editing, L.-S.C.; visualization, R.G.M. and Y.-Y.W.; supervision, L.-S.C.; project administration, L.-S.C.; funding acquisition, L.-S.C. All authors have read and agreed to the published version of the manuscript.

**Funding:** This research was funded in part by the National Science and Technology Council, Taiwan (grant no. MOST 111-2410-H-324-006).

**Institutional Review Board Statement:** Not applicable.

**Informed Consent Statement:** Not applicable.

**Data Availability Statement:** The used secondary data that were used for analysis in this study are available from the corresponding author upon request.

**Conflicts of Interest:** The authors declare no conflict of interest.

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
