# Peer review of "Analyzing Online Fake News Using Latent Semantic Analysis: Case of USA Election Campaign"

_2504-2289, doi:10.3390/bdcc7020081_

Round 1

Reviewer 1 Report

The authors aim to develop an effective method which combines NLP and LSA using SVD techniques to help social scientists to analyze fake news for discovering the exact elements. The authors made some spelling mistakes. Furthermore, some parts are not well understood from the document. My comments are as follows:

1. The author made some spelling mistakes in the abstract. For example, in the fifth line, Uulike should be changed to Unlike. In line 9, for example, analyse should be revised as analyze. Please revise all spelling mistakes in this paper.

2. In my opinion, the explanation of relevant concepts does not need to be detailed in the Related works, which only needs to explain the research progress on the identification of true and fake news.

3. The abbreviation for Natural Language Processing has been explained in the Introduction and need not be repeated in Methodology Step2.

4. Some of the writing in the manuscript is irregular. For example, ti and dj in Step 2.5, t and d are in italics in the formula (1), and i is in italics as subscript, but not in the form of italics and subscript in the explanation. A similar problem occurs in Step 3.1. Please revise all irregular writing in this paper.

5. The authors conduct singular value decomposition for matrix A in Step3.1, but what matrix A is is not explained, please add.

6. The authors construct a 5009*10019 TDM with TF-IDF weights for LSA. I'd like to know the basis for choosing 5009*10019.

7. In the conclusion part, the authors expound the impact of fake news in disruption information era for current situation at a large length. I think these explanations and effects should appear in the introduction part.

Reviewer 2 Report

The paper discusses the problem of fake news and its impact on society. It highlights that while there are several studies on fake news detection and sentiment analysis, few researchers have paid attention to analyzing fake news in-depth. The article proposes a method that combines natural language processing and Latent Semantic Analysis using Singular Value Decomposition techniques to help social scientists analyze fake news and discover its exact elements. The article further mentions that the study aims to analyze fake political news, which is a significant problem.

The methodology section discusses the issue of fake news and the limited attention given to fake news analysis, particularly in the context of politics. The author notes that most previous studies have focused on detecting and identifying fake news and analyzing fake news sentiment, using machine learning or deep learning techniques. However, this study aims to develop an effective method for analyzing fake news that combines natural language processing and Latent Semantic Analysis using Singular Value Decomposition  techniques, with the goal of discovering the exact elements that make news fake. The review also notes that the study will analyze the characteristics of both true and fake news and use a real case from the USA Election Campaign in 2016 to demonstrate the effectiveness of the proposed method. 

The method is demonstrated using a real case from the USA Election Campaign in 2016. The article describes the step-by-step process of data collection, pre-processing, tokenization, data cleaning, lemmatization, and count word frequency to create a term-document matrix using the TF-IDF weights. The TDM is then subjected to LSA using SVD, and the results are reduced in dimensional space using the Scree Test. The orthogonal rotation of axes is then performed to extract the concept load LT and each feature word is ranked according to the load to name the word concept. The named concepts are then analyzed to understand and define each concept for accurate and fake news analysis. The article's approach is different from other published works as it focuses on analyzing fake news rather than detecting fake news. The article provides useful suggestions for future researchers to distinguish fake news.

The authors define five concepts of fake news that were employed during the 2016 US election campaign to manipulate readers. They also present a table summarizing the differences between fake news and real news, including their sources, viewpoints, and domains. The article then provides a table of criteria for newsworthiness from a journalism perspective, which both fake and accurate news can contain. The authors analyze the structure of both fake and real news and find that they have a complete structure, including a title, lead, and data section. The article also discusses the concept of manipulation and delayed facts in journalistic practice, highlighting the role of social media in disseminating fake news. The authors argue that news culture in the media industry should be reimagined to combat fake news and that delayed facts are a common feature of online news.

The article's strength is that it addresses an important issue of fake news and proposes a method to analyze it. The article is relevant in the current context as fake news is becoming increasingly prevalent, and there is a need for effective methods to combat it. The article's focus on analyzing fake political news is also commendable, as this type of fake news can significantly impact democracy and society.

One potential disadvantage of this design is that the method is limited to analyzing election fake news. While this is an important topic, the method may not be applicable to other types of fake news, such as health-related or financial news. Additionally, the method relies on machine learning algorithms, which may be biased or limited in their ability to detect complex nuances in language. It is important for the researchers to address these limitations and ensure that the method produces accurate results.

Dataset Bias: The study relies on a specific dataset of online news data collected for the purpose of examining fake news in the context of election campaigns. This dataset may not be representative of other types of fake news or topics, and the results of the study may not be generalizable to other contexts.

SVD Limitations: The Singular Value Decomposition (SVD) technique used in the study for Latent Semantic Analysis (LSA) has some limitations in terms of scalability and computational efficiency. The study does not address these potential limitations and their impact on the analysis.

Reviewer 3 Report

This article applies Latent Semantic Analysis (LSA) to analyze fake news contents. While most research tackles the problem of classifying fake news against accurate news, the authors focus on identifying the major elements in fake news, and detected five major concepts from a fake news corpus sourced during the 2016 U.S. presidential election. The authors describe in detail how they extract these concepts and summarize them based on key words. They then discuss some major differences between fake news and accurate news.

Before I comment on the content of the paper and the value of its findings, I have to first express my concern on the paper’s writing. While grammar issues do not affect the overall quality of the paper and can be easily fixed by editing, I feel the paper’s writing has major difficulties in conveying the authors’ intentions. There are many cases where a sentence doesn’t really make sense, or a sentence contradicts with the previous sentence. Occasionally, a paragraph is so vague that it is very hard to understand the authors’ argument. I will provide a few examples below:

  • Section 1, paragraph 3: “No one can avoid online news becoming a solution, or even a problem at the same time.” I’m confused by what this sentence is trying to say.
  • Section 2.1, first paragraph: It first says “Journalism is an ideology about news and how facts are discovered and reported.”. However, the next sentence says “Ideologically, from a journalism perspective, it has at least four ideologies”. How many ideologies are actually there?
  • Last paragraph in section 5: the last few sentences are: “We can think of the term delayed facts as an advantage or disadvantage of online journalism. However, further discussions can be made from any point of view. However, things backfired, and things backfired. The influence of public opinion at the beginning of the news has its weaknesses.” I find this part to be extremely obscure and I really can’t make sense of what it is trying to say.

The grammatical issues, while not a big concern, are also very prevalent in this article. For example, there are two grammar errors in the abstract: “fake news always produced” should be “fake news are always produced”, and “it spread” should be “it spreads”. I would recommend the authors to use an English editing service (like Grammarly) before submitting their article the next time.

I appreciate the authors’ interest in understanding the elements of fake news, instead of simply detecting fake news. A minor comment here: the abstract uses the phrases “fake news analysis” or “analyst fake news”, which are rather vague. It is not until much later that the readers find that they mean identifying major elements and concepts in fake news. It would be great if the article clarifies this at the beginning. 

I think the article’s usage of LSA (i.e. applying SVD to a TFIDF matrix) is sound. However, although few works have applied this method to fake news corpora, using LSA to create latent topic vectors is a very well-established approach and has been popular since the 1990s. Therefore, I think while the article has some novelty value, it is quite limited. The article also uses a Kaggle dataset, so there is no added value to the research community by publishing a new dataset.

Regarding the paper’s main results, I think the authors do a good job in identifying the most important eigenvectors using a scree plot. However, it looks like there is not a clear separation between Concept #1 and Concept #2 that they identify. For example, both concepts include the keywords “political” and “senate”, and it is not clear to me why they name the first concept “coalition” in particular. 

In Table 4, the article lists several major differences between fake and factual news. They look quite reasonable, but it appears they’re not data driven as the LSA is only applied to the fake news dataset. If the authors seek to further improve this work, I think it will be very interesting to apply the methods to the factual news dataset and make a comparison.

One last comment: the article makes references to the fake news situation in several countries, including Indonesia, Lebanon, Kenya, etc. However, each of these occurrences seems a bit random and are not connected to the whole picture. It would be great if the article includes further motivation of why these countries are selected as examples.

To conclude, I think while the article tackles an interesting problem and has sound methodology, it unfortunately doesn’t add a lot of value to its research domain. Combined with the worrisome quality of writing, it is not a good fit for publication in its current status.

Reviewer 4 Report

In the abstract, more emphasis should be placed on methodology and results.

Introduction and relatet wor are meaningful and describe the prerequisites for the studies sufficiently.

Methodology

The technical parameters of the implementation are missing here: hardware, software, computing performance, etc.

Parts of the relevant methodological information can be found in 4.1. these should be integrated into 3.

Results discusion and Conclution

table 5 has two columns with exactly the same content. this comparison does not add any value for the reader.

The embedding of tables 4, 5 and 6 in the text should be improved.

the conclusion is very long, which is unusual. the authors should integrate the main contents into the discussion and only present the statements of the scientific work in a few sentences. parts of the first section of the discussion could be integrated into the resualts section.

other comments:

The authors very often use active formulations such as we - Here, care should be taken that the scientific writing is passive or that formulations such as in the study presented etc. are used.

Round 2

Reviewer 1 Report

The authors have improved the manuscript according to the reviewer's comments.

Author Response

Thank you very much for your comments and suggestions. Proof-reading has been done by a native English speaker.

Reviewer 3 Report

I appreciate the authors' work on revising this work, and I think the paper's quality has been improved considerably. I'm still a little concerned about the quality of writing though. For example, there is a grammar error in the first sentence of the abstract; it uses a plural form for fake news with "fake news are", but then used a singular form with "it spreads". I hope the editor can work with the authors to fix the issues with writing before publication.

Author Response

Many thanks for your kind efforts and suggestions. This manuscript has been edited by a native English speaker. By the way, we also correct the citation format according to the authors' guide of BDCC.

Reviewer 4 Report

the authors have implemented the comments very well. in my view, the current version is in order and can be published.

Author Response

Thank you very much for your efforts and suggestions.